# From Translation to Multilinguality: Revisit the Role of Parallel Data in Multilingual LLM Pretraining

## Abstract

Multilingual large language models (MLLMs) are commonly trained with parallel data (i.e., concatenated translation pairs) to introduce cross-lingual alignment signal and induce capabilities transfer for non-English languages. However, it remains unclear whether this de facto practice improves general multilingual ability beyond translation. We conduct a controlled, large-scale study comparing two ways of using parallel data in pretraining: (1) standard concatenated translation pairs as a single sample, and (2) treating each side as an independent sample. Across diverse experimental settings, we find consistent results: parallel concatenation yields substantial gains on translation metrics, but offers limited benefits for general monolingual abilities and cross-lingual abilities. This result suggests that while parallel-form alignment signals directly build translation ability, they do not readily transfer into broader multilingual competence through standard learning process. Motivated by this gap, we propose a pragmatic multi-step pipeline to leverage the translation ability induced by parallel data in a data-driven perspective, which consistently improves general monolingual and cross-lingual performance. Our findings clarify the role and limits of parallel data in MLLM pretraining and offer a practical recipe for building more comprehensively capable multilingual models.

## 1 Introduction

Multilingual large language models (MLLMs) (Le Scao et al., 2023; Üstün et al., 2024; Wei et al., 2023) underpin a growing range of applications, from global information access and cross-lingual retrieval to translation, multilingual assistants, and knowledge transfer across linguistic communities. Yet, with high-resource languages (notably English) dominating the web-scale training mixture, the pretraining of MLLMs remains fundamentally constrained by the scarcity and imbalance of low-resource data. A prevailing industry practice seeks to mitigate the shortfall by injecting cross-lingual alignment signals through parallel data (Qorib et al., 2025; Reid & Artetxe, 2022; Kale et al., 2021)—concatenated translation pairs of semantically equivalent content from two languages, typically English and a low-resource language, within a single training sample.

The formation of parallel data is intuitively appealing: by co-presenting aligned semantics across languages, it appears to offer a direct supervision signal for the model to form shared representations between languages, which may help to transfer capabilities from high-resource to low-resource languages, and to complete cross-lingual tasks (Qorib et al., 2025; Cheng et al., 2025). Despite its widespread adoption, however, the extent to which this concatenation-based use of parallel data enhances general multilingual ability beyond translation remains an underexplored empirical question.

In this paper, we investigate the role and limitations of parallel data in MLLM pretraining. We pose a focused research question: How does the concatenation format of parallel data influence the multilingual capabilities of MLLMs? More concretely, does training on concatenated translation pairs improve general monolingual abilities and cross-lingual abilities such as reasoning and understanding that go beyond translation?

To answer these questions, we conduct a controlled study that isolates the effect of the parallel-data format in pretraining. We systematically compare two protocols: (1) the standard practice of concatenating translation pairs into a single sample, and (2) treating each side of the translation pair as an independent sample within the pretraining corpus. We undertake extensive and comprehensive data preparation covering 18 languages, encompassing both monolingual and cross-lingual training corpora as well as evaluation sets that probe a broad spectrum of multilingual abilities. On this foundation, we establish concise baselines and benchmarks. To ensure the robustness and generality of our findings, we vary a diverse set of experimental settings, including choices of language combination (English plus single non-English language with varying linguistic similarity, and English plus multiple co-existing non-English languages), the proportion of parallel data, whether to use English-then-multilingual pretraining schedules, and different model scales.

Our experiments yield clear and consistent results, shedding light on the effect and limits of parallel data in multilingual pretraining. Based on the results, we also propose a practical approach that leverages the benefits of parallel data to achieve broader improvements in multilingual capabilities. Taken together, these results provide concrete guidance for turning parallel data into multilingual gains in LLM pretraining practice. In summary, this work makes three contributions:

- We provide a systematic, controlled comparison between standard concatenated parallel data and split (independent) translation pairs across a wide range of pretraining configurations.

- We crystallize a key insight: directly using concatenated parallel data primarily improves translation ability, but does not significantly enhance general monolingual competence or cross-lingual ability.

- We propose and validate a practical approach that takes advantage of the translation ability obtained from parallel data, yielding robust gains in both monolingual and cross-lingual performance.

## 2   RELATED WORK

### 2.1   MULTILINGUAL LARGE LANGUAGE MODELS

The emergence of Large Language Models (LLMs) (Achiam et al., 2023; Touvron et al., 2023; Jiang et al., 2023) has profoundly reshaped the landscape of Natural Language Processing (NLP). Although initial breakthroughs were primarily in English (Biderman et al., 2023; Groeneveld et al., 2024), a substantial research effort has since focused on extending these capabilities to multiple languages, leading to the development of Multilingual Large Language Models (MLLMs) (Le Scao et al., 2023; Üstün et al., 2024; Wei et al., 2023). Following the success of its monolingual predecessor, the first major multilingual model, multilingual BERT (mBERT) (Devlin et al., 2019), was introduced. It adapted the BERT training procedure to a massive dataset of Wikipedia text in 104 languages. This breakthrough paved the way for a new generation of multilingual large language models, including XLM-R (Conneau et al., 2019), mBART (Liu et al., 2020), and mT5 (Xue et al., 2020). Over time, larger models like PaLM (Chowdhery et al., 2023), BLOOM (Le Scao et al., 2023), and LLaMA (Touvron et al., 2023) have been developed to achieve state-of-the-art results on complex, multi-step reasoning tasks in multiple languages.

Nezhad & Agrawal (2024) investigated how various factors affect the performance of multilingual large language models. For seen languages, the most significant factor influencing performance is the pretraining data size. For unseen languages, script type and language family are the most crucial factors. The study also found that model size and architecture had little impact on these key findings, offering insights for building more effective multilingual NLP systems. To understand how LLMs process multilingual text and their underlying mechanisms, Tang et al. (2024) proposes a new method called Language Activation Probability Entropy (LAPE) to pinpoint language-specific neurons within LLMs. Using LAPE, the researchers found that an LLM's proficiency in a specific language is largely due to a small group of neurons, mainly located in the model's top and bottom layers.

### 2.2   PARALLEL DATA

The pretraining of Multilingual Large Language Models (MLLMs) is profoundly influenced by the data they are trained on. A widely held belief in the field is that parallel data, which consists of text aligned across two or more languages, is crucial for developing strong machine translation and cross-lingual understanding capabilities (Qorib et al., 2025; Reid & Artetxe, 2022; Kale et al., 2021). Prior work has explored the nuances of this relationship. For example, Reid & Artetxe (2022) utilize unsupervised machine translation to create synthetic parallel data. Their research reveals that even generated parallel data can improve multilingual performance on downstream tasks. Kale et al. (2021) examines the effect of including parallel data during the pretraining of mT5. The findings show that adding tasks like machine translation to the pretraining process is a simple and effective way to boost performance on various multilingual and cross-lingual tasks. However, we note that contracting training with and without parallel data introduces discrepancy of data in the content level, potentially confounding the real effect of parallel formation.

Additionally, the research by Qorib et al. (2025) systematically investigates the effect of including parallel data on large language models' multilingual capabilities, specifically on translation and multilingual common-sense reasoning. However, their experiments only focus on English, Chinese and Indonesian, and evaluate only mono-lingual performance, which makes the results somewhat limited. We expand on previous research by including more languages, conducting more detailed and equitable comparisons under various settings, and performing more comprehensive testing.

# 3 INVESTIGATING THE EFFECT OF PARALLEL DATA

## 3.1 DATA PREPAREATION

We perform extensive data curation for the experiments. All of our data are derived from web-collected sources (e.g., Common Crawl, CCAligned (El-Kishky et al., 2019), CCMatrix (Schwenk et al., 2019), ParaCrawl (Bañón et al., 2020)) and undergo an pipeline of parsing and filtering. We apply a language classifier to label all texts and retain 18 languages, including English. To approximate real-world MLLM pretraining while keeping the setup concise, we intentionally exclude data sources that are common in practice but extraneous to our research objective (e.g. code and math data). For parallel data, we retain only translation pairs that has an English text part. When using standard parallel data, following Cheng et al. (2025), the text pair is concatenated with an explicit language tag (e.g., `<en>`) to indicate the boundary and identity of both segments. The order of the pair is randomized.

## 3.2 PROTOCOLS

To investigate the effect of parallel data in multilingual LLM pretraining, we conduct systematical experiments with a focus on its featuring format. Specifically, we compare two ways of incorporating parallel data in training dataset:

(1) **Standard**: Adopt the standard parallel-format data (i.e., concatenated translation pairs). When trained on a parallel sample, a model has access to the translation counterpart through its context window, which provides explicit cross-lingual alignment signal.

(2) **Split**: Split the pair, and treat the text of each side as an independent sample. In this way, each split part is analogue to a normal monolingual sample, offering no direct cross-lingual supervision.

## 3.3 EXPERIMENT SETTINGS

The training data comprises exactly three components: (i) English monolingual data, (ii) non-English monolingual data, and (iii) parallel data (or the split version). To ensure the generality of our findings, we compare the two protocols for using parallel data across varying settings on multiple dimensions. Specifically, we explore the following axes:

- **Setting I: English plus single non-English language.** In this setting, each controlled comparison constructs training data from English plus one non-English language. Based on linguistic similarity to English, we carefully select three languages: German, French, and Japanese. German is most similar to English as both belong to the Germanic branch; French and English are from different branches but within the Indo-European family; Japanese belongs to a totally different language family. For each of these languages, we contrast pretraining with standard concatenated parallel data versus split parallel data, thereby examining how the effectiveness of parallel data varies with different degrees of similarity to English.

- **Setting II: English plus multiple non-English languages.** In this setting, we train with English plus all 17 non-English languages we have collected, covering a spectrum of linguistic similarity and variability in richness of data resource. Non-English monolingual data comprises 36% of training tokens while parallel data comprises 14% share. See figure 1 and figure 2 for non-English languages we use and their relative proportion. This enables a more holistic assessment of how parallel data behaves when many languages co-exist in the corpus.

- **Setting III: Proportion of parallel data.** In comparison to the 14% share of parallel data in Setting II, in this setting, we increase the parallel proportion to 20% and decrease it to 8% to evaluate whether the effectiveness of parallel data is sensitive to changes in proportion. Given the limited availability of parallel data, this covers the typical range of parallel proportion in practice.

- **Setting IV: Two-stage multilingual training.** Some prior studies (Qorib et al., 2025; Cheng et al., 2025) adopt a two-stage curriculum that pretrains on English first and then on multilingual data. We replicate this setup: Stage 1 trains on English-only data; Stage 2 trains on multilingual monolingual data and parallel data under both protocols. We then measure whether the effect of parallel data remains consistent in this curriculum.

- **Setting V: Model scale.** Experiments in Setting I to IV use relatively small-sized 1.5B-parameter model due to computational resource limitation. To examine whether model scale alters the conclusions, we conduct additional comparison with an 8B-parameter model to assess the impact of parallel data at larger scales.

**Implementation Details** Due to the scarcity of available data for single non-English language, experiments in Setting I are trained with 100B tokens, others with 300B tokens. For the same reason, parallel data comprises

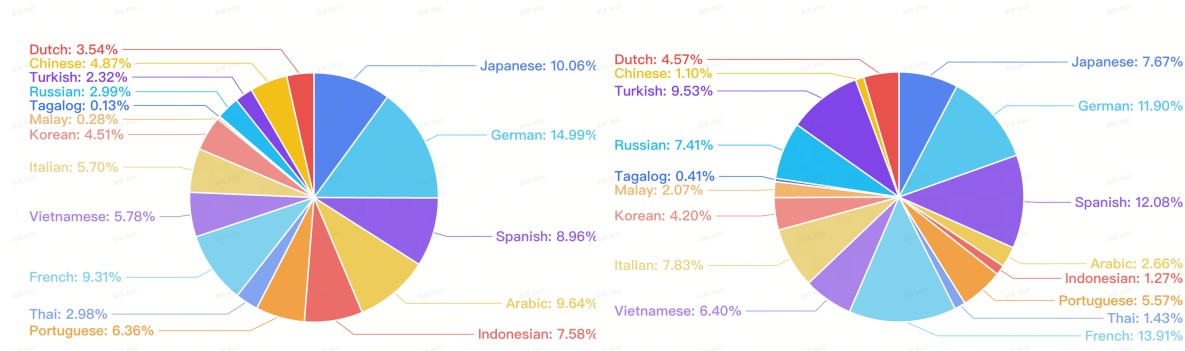

Figure 1: Relative proportion for non-English languages in monolingual data.

Figure 2: Relative proportion for non-English languages in parallel data.

| Protocol | General Monolingual Ability | | | | | | | Translation Ability | | | |
|---|---|---|---|---|---|---|---|---|---|---|---|
| | HellaSwag | ArcC | ArcE | MMLU | XNLI | Avg | Δ | EN→XX | XX→EN | Avg | Δ |
| **Setting I: English + German** | | | | | | | | | | | |
| Standard | 44.47 | 30.72 | 58.38 | 30.78 | 45.66 | 42.00 | +0.16 | 53.69 | 61.23 | 57.46 | +13.28 |
| Split | 44.08 | 30.46 | 58.33 | 30.03 | 46.31 | 41.84 | | 39.70 | 48.65 | 44.18 | |
| **Setting I: English + French** | | | | | | | | | | | |
| Standard | 50.72 | 33.70 | 60.19 | 30.64 | 47.51 | 44.55 | +0.31 | 60.94 | 61.62 | 61.28 | +13.70 |
| Split | 50.37 | 32.59 | 60.48 | 30.34 | 47.43 | 44.24 | | 42.63 | 52.36 | 47.50 | |
| **Setting I: English + Japanese** | | | | | | | | | | | |
| Standard | 39.45 | 28.92 | 50.34 | 27.72 | - | 36.61 | -0.33 | 31.90 | 44.67 | 38.29 | +16.13 |
| Split | 39.76 | 29.18 | 50.97 | 27.85 | - | 36.94 | | 19.19 | 25.12 | 22.16 | |

Table 1: Evaluation results of experiments in Setting I. We use 10-shot test for Hellaswag, 25-shot for ArcChallenge and ArcEasy, 5-shot for MMLU, XNLI and FloresTranlate. We report normalized accuracy (acc_norm) percentage for HellaSwag, ArcChallenge, ArcEasy and MMLU, accuracy percentage for XNLI, and CHRF score for FloresTranslate. For MMLU, we evaluate with the lighteval variant (Alzahrani et al., 2024) for better consistency. Δ column shows the difference between the average performance of two protocols (Standard relative to Split).

10% of training data in Setting I, and 14% in Setting II, IV and V. In all experiments, English monolingual data comprises 50% of training datasets. The rest share is left for non-English monolingual data. The order of samples in training dataset is randomly shuffled. For the 1.5B model, we use a 24-layer Llama-structured transformer (Touvron et al., 2023). The 8B model has the same structure, with 32 layers and higher hidden dimension.

## 3.4 EVALUATION

To comprehensively assess multilingual capability, we group evaluations into three categories: general monolingual ability, general cross-lingual ability and translation ability. For experiments in Setting I, we evaluate only the target language; for all other settings, we evaluate across the full set of 17 languages.

**General monolingual ability** These benchmarks probe general abilities within a single non-English language (e.g., natural language understanding, commonsense reasoning, technical reasoning, etc.). Performance on these benchmarks reflect whether parallel data help with ability transfer from high-resource language into low-resource ones. We evaluate on HellaSwag (Zellers et al., 2019), ARC-Easy (Clark et al., 2018), ARC-Challenge (Clark et al., 2018), MMLU (Hendrycks et al., 2009), XNLI (Conneau et al., 2018) and XStoryCloze (Lin et al., 2022). XNLI and XStoryCloze do not support all languages we used. In that case, we report the results on covered languages.

**General cross-lingual ability** Cross-lingual ability refers to the ability to complete tasks in a required language other than that of the context. We evaluates experiments in Setting II V on the cross-lingual benchmarks provided by MuBench (Han et al., 2025) that pair English with an non-English language, including cross-lingual version of

| Protocol | General Monolingual Ability | | | | | | | |
|---|---|---|---|---|---|---|---|---|
| | HellaSwag | ArcC | ArcE | MMLU | XNLI | XStoryCloze | Avg | Δ |
| **Setting II: English plus multiple non-English languages** | | | | | | | | |
| Standard | 45.03 | 32.15 | 56.37 | 30.19 | 42.10 | 57.75 | 43.93 | +0.15 |
| Split | 44.43 | 32.40 | 56.03 | 30.23 | 42.07 | 57.51 | 43.78 | |
| **Setting III: Proportion of parallel data (8%)** | | | | | | | | |
| Standard | 45.36 | 33.27 | 56.90 | 30.42 | 42.83 | 57.74 | 44.42 | +0.32 |
| Split | 44.85 | 32.70 | 56.11 | 30.35 | 43.21 | 57.40 | 44.10 | |
| **Setting III: Proportion of parallel data (20%)** | | | | | | | | |
| Standard | 44.52 | 32.29 | 56.03 | 30.25 | 41.89 | 57.64 | 43.77 | +0.12 |
| Split | 43.92 | 32.35 | 55.97 | 29.98 | 42.26 | 57.40 | 43.65 | |
| **Setting IV: Two-stage multilingual training** | | | | | | | | |
| Standard | 43.37 | 30.83 | 54.08 | 29.05 | 43.22 | 57.43 | 43.00 | +0.28 |
| Split | 42.90 | 30.62 | 53.57 | 28.95 | 43.29 | 57.00 | 42.72 | |
| **Setting V: 8B-parameter model scale** | | | | | | | | |
| Standard | 53.19 | 41.85 | 66.43 | 34.50 | 43.61 | 61.64 | 50.20 | +0.38 |
| Split | 53.33 | 40.87 | 65.35 | 34.29 | 43.64 | 61.44 | 49.82 | |

| Protocol | General Cross-lingual Ability | | | | | | | Translation Ability | | | |
|---|---|---|---|---|---|---|---|---|---|---|---|
| | HellaSwag | ArcC | ArcE | MMLU | StoryCloze | Avg | Δ | EN→XX | XX→EN | Avg | Δ |
| **Setting II: English plus multiple non-English languages** | | | | | | | | | | | |
| Standard | 43.38 | 30.24 | 43.45 | 27.12 | 59.60 | 40.76 | -0.54 | 47.98 | 58.32 | 53.15 | +8.78 |
| Split | 43.82 | 29.55 | 45.53 | 27.39 | 60.22 | 41.30 | | 38.51 | 50.22 | 44.36 | |
| **Setting III: Proportion of parallel data (8%)** | | | | | | | | | | | |
| Standard | 44.06 | 30.93 | 44.93 | 27.52 | 59.52 | 41.39 | +0.12 | 47.45 | 57.56 | 52.51 | +7.92 |
| Split | 43.94 | 30.76 | 43.45 | 27.61 | 60.60 | 41.27 | | 38.35 | 50.82 | 44.59 | |
| **Setting III: Proportion of parallel data (20%)** | | | | | | | | | | | |
| Standard | 43.87 | 30.84 | 44.72 | 27.25 | 60.84 | 41.50 | -0.21 | 48.72 | 58.53 | 53.63 | +10.10 |
| Split | 43.86 | 30.33 | 46.21 | 27.60 | 60.53 | 41.71 | | 37.69 | 49.35 | 43.52 | |
| **Setting IV: Two-stage multilingual training** | | | | | | | | | | | |
| Standard | 40.48 | 27.66 | 40.31 | 27.02 | 58.82 | 38.86 | -0.08 | 49.02 | 58.14 | 53.58 | +9.15 |
| Split | 40.61 | 27.06 | 41.50 | 27.03 | 58.51 | 38.94 | | 39.57 | 49.28 | 44.43 | |
| **Setting V: 8B-parameter model scale** | | | | | | | | | | | |
| Standard | 51.92 | 35.05 | 51.67 | 28.85 | 64.78 | 46.45 | +0.06 | 52.31 | 61.76 | 57.03 | +5.87 |
| Split | 51.98 | 33.59 | 51.67 | 29.41 | 65.33 | 46.40 | | 45.42 | 56.90 | 51.16 | |

Table 2: Evaluation results of Setting II to V. We report 0-shot accuracy for XStoryCloze, 5-shot normalized accuracy for cross-lingual StoryCloze. Metrics used for other cross-lingual benchmarks are the same as their corresponding monolingual ones. Performance is averaged over 17 languages.

from HellaSwag, ARC-Easy, ARC-Challenge, StoryCloze, and MMLU. Experiments in Setting I is not evaluated for the lack of dedicated benchmarks.

**Translation ability** Though translation can be thought of as a special case of cross-lingual task, we consider it separately for its direct relation to the formation of parallel data. We evaluate on FLORES-Translate (Goyal et al., 2022), covering both translating directions: English to non-English and the opposite.

## 3.5 RESULTS

Table 1 reports results for experiments combining English with a single non-English language. Across all three target languages, the Standard protocol yields substantial gains in bidirectional translation performance relative to

| Experiment | General Monolingual Ability | | | | | | | |
|---|---|---|---|---|---|---|---|---|
| | HellaSwag | ArcC | ArcE | MMLU | XNLI | XStoryCloze | Avg | $\Delta$ |
| Baseline | 53.19 | 41.85 | 66.43 | 34.50 | 43.61 | 61.64 | 50.20 | +1.34 |
| Proposed | 56.20 | 43.53 | 68.40 | 35.62 | 43.85 | 61.61 | 51.54 | |

| Experiment | General Cross-lingual Ability | | | | | | | Translation Ability | | | |
|---|---|---|---|---|---|---|---|---|---|---|---|
| | HellaSwag | ArcC | ArcE | MMLU | StoryCloze | Avg | $\Delta$ | EN→XX | XX→EN | Avg | $\Delta$ |
| Baseline | 51.92 | 35.05 | 51.67 | 28.85 | 64.78 | 46.45 | +2.93 | 52.31 | 61.76 | 57.03 | +0.84 |
| Proposed | 55.69 | 38.49 | 54.35 | 29.41 | 68.96 | 49.38 | | 53.26 | 62.49 | 57.87 | |

Table 3: Evaluation results of proposed approach. Both experiments are under standard protocol. $\Delta$ column shows the average performance gains of proposed approach.

the Split protocol, with CHRF score improvements over 10 points. This aligns with intuition, as the construction of parallel data directly targets translation objectives.

In contrast, general monolingual abilities in the three languages do not exhibit notable improvements. The average difference between the two protocols remains within approximately +0.3%, with no clear correlation to linguistic similarity with English (French shows slightly larger gains than the other two languages, yet its similarity to English lies between them). These findings indicate that the model does not trivially leverage the cross-lingual alignment signal in parallel data to transfer the dominant English competence into stronger monolingual abilities for low-resource languages.

Building on the insights from Setting I experiments, Settings II–V expand to the full set of languages and incorporate evaluations beyond translation to assess general cross-lingual abilities. The results are summarized in Table 2. Once again, we observe that the standard parallel data yields substantial improvements in translation performance relative to split version. However, regardless of whether we vary the proportion of parallel data, adopt a two-stage training schedule, or scale the model up to 8B parameters, the gains in general monolingual ability remain limited, with the largest improvement being +0.38%. Moreover, we find no clear advantage of the standard protocol over the split protocol on general cross-lingual abilities under any configuration, in stark contrast to the consistent improvement on translation benchmark.

Putting it altogether, these results clearly reveal the effect and limitations of parallel data in multilingual pre-training: they chiefly improve performance on translation tasks, with no remarkable contribution to other general multilingual tasks (neither monolingual nor cross-lingual). This reflects the dominant influence of data formation in model learning: the concatenated format of parallel data aligns directly with the translation objective and therefore has an immediate effect. In contrast, the relationship between general multilingual tasks and translation ability is indirect. Under the standard training process, the model does not readily achieve such capability transfer.

## 4 A PRACTICAL APPROACH LEVERAGING PARALLEL DATA

Building on the above results and analyses, in this section, we propose a practical approach that leverages parallel data to enhance general multilingual capabilities beyond translation. Guided by insights into the data-driven nature of model learning, we adopt a staged pipeline:

- **Step 1**: Train a strong translator. We first train a model with parallel data to develop strong translation capability and then, with lightweight post-training, derive a translation specialist model.
- **Step 2**: Synthesize ability-oriented multilingual corpora from English data with the specialist model. Specifically, for general monolingual ability, we use the specialist to synthesize non-English monolingual data; for general cross-lingual ability, we synthesize code-switch data (Wang et al., 2025) by replace each English sentence with its translation with 50% probability; for translation, we synthesize parallel data by concatenating a English sentence and its translated version with language tag.
- **Step 3**: Integrate synthesized data into pretraining. We add the synthesized corpora to the training mix, enabling the new model to acquire the corresponding capabilities.

**Experiment setup**  We evaluate this approach with the standard-protocol 8B model from Setting V as a baseline. We post-train the model on 100,000 instances from our parallel data plus public FLORES development set (Goyal et al., 2022) and OpenHermes (Teknium, 2023), yielding the translation model. From the baseline's English corpus, we randomly sample a portion as the source for translation, generating non-English monolingual data,

| Experiment | General Monolingual Ability | | | | | | | |
|---|---|---|---|---|---|---|---|---|
| | HellaSwag | ArcC | ArcE | MMLU | XNLI | XStoryCloze | Avg | Δ |
| Baseline | 44.43 | 32.40 | 56.03 | 30.23 | 42.07 | 57.51 | 43.78 | -0.04 |
| Unpaired | 44.21 | 32.31 | 55.79 | 30.15 | 43.16 | 57.30 | 43.82 | |

| Experiment | General Cross-lingual Ability | | | | | | | Translation Ability | | | |
|---|---|---|---|---|---|---|---|---|---|---|---|
| | HellaSwag | ArcC | ArcE | MMLU | StoryCloze | Avg | Δ | EN→XX | XX→EN | Avg | Δ |
| Baseline | 43.82 | 29.55 | 45.53 | 27.39 | 60.22 | 41.30 | -0.12 | 38.51 | 50.22 | 44.36 | +0.28 |
| Unpaired | 43.44 | 31.36 | 45.06 | 27.85 | 59.37 | 41.42 | | 38.12 | 50.05 | 44.08 | |

Table 4: Evaluation results of unpaired data experiment. Δ column shows the average performance difference (Baseline relative to Unpaired).

| Protocol | General Monolingual Ability | | | | | | |
|---|---|---|---|---|---|---|---|
| | HellaSwag | ArcC | ArcE | MMLU | XNLI | XStoryCloze | Avg |
| Standard | 45.03 | 32.15 | 56.37 | 30.19 | 42.10 | 57.75 | 43.93 |
| Split | 44.43 | 32.40 | 56.03 | 30.23 | 42.07 | 57.51 | 43.78 |
| Discard | 44.21 | 32.31 | 55.79 | 30.15 | 43.16 | 57.30 | 43.82 |

| Protocol | General Cross-lingual Ability | | | | | | Translation Ability | | |
|---|---|---|---|---|---|---|---|---|---|
| | HellaSwag | ArcC | ArcE | MMLU | StoryCloze | Avg | EN→XX | XX→EN | Avg |
| Standard | 43.38 | 30.24 | 43.45 | 27.12 | 59.60 | 40.76 | 47.98 | 58.32 | 53.15 |
| Split | 43.82 | 29.55 | 45.53 | 27.39 | 60.22 | 41.30 | 38.51 | 50.22 | 44.36 |
| Discard | 43.44 | 31.36 | 45.06 | 27.85 | 59.37 | 41.42 | 38.12 | 50.05 | 44.08 |

Table 5: Evaluation results of discarding parallel data.

code-switch data and parallel data as described in step 2. We then replace one-third of the baseline's non-English data with synthesized monolingual data and code-switch data respectively, and replace baseline's parallel data with synthesized parallel data, keeping the total data volume unchanged. Relative proportions between languages and all other experimental configurations remain identical to the baseline.

**Results** Table 3 reports the results of the proposed approach compared to the baseline. The method delivers broad improvements in multilingual capability: gains in general monolingual ability and general cross-lingual ability are markedly higher than those achieved by the use of standard parallel data itself (if any), and translation performance further improves on top of the already strong baseline under standard protocol. These results demonstrate the effectiveness of leveraging translation capability obtained from parallel corpora to enhance general multilingual performance in a data-driven paradigm.

**Discussion** Note that while introducing translated data incurs some additional cost, this cost is one-off: the synthesized datasets can be reused in subsequent training runs. In practice, the efficacy of this approach also helps alleviate the scarcity of low-resource language data (both monolingual and parallel), and has the potential to extend to other data format and contribute to associated multilingual tasks.

## 5 ABLATION STUDY

### 5.1 UNPAIRED DATA

Under the split protocol, the two sides of a translation pair are used as independent samples. Although the model cannot access explicit alignment signals within the same training context, it could, in principle, still implicitly associate the two sides across samples. To ablate this potential effect, we conduct an unpaired data experiment.

Using the Setting II split-protocol experiment as a baseline, we randomly retain half of the English samples obtained from the split parallel corpus and, for the non-English side, exclude the corresponding paired half. This

yields an unpaired dataset. We replace the baseline's split parallel data with this unpaired corpus and upsample it by 2× to keep the total token count unchanged. All other experimental configurations are identical to the baseline.

Results are shown in Table 4. We observe that the unpaired setup performs similarly to the baseline across all multilingual abilities. This indicates that, under the split protocol, the model does not implicitly leverage independent translation samples in the data to enhance multilingual capability.

## 5.2 DISCARDING PARALLEL DATA

Some prior studies (Qorib et al., 2025; Reid & Artetxe, 2022; Kale et al., 2021) contrasted training with versus without parallel data. Though straightforward, such comparisons can be confounded by differences in content distribution between parallel and non-parallel sources (quality, diversity, language mix, etc.). Our primary protocol differs: we compare parallel data as a special-format sample (concatenated translation pairs) against splitting the pairs into ordinary samples. We argue that this design better controls content distribution and avoids introducing new variables.

Nevertheless, we additionally conduct a direct "Discard" protocol experiment that removes parallel data. Building on Setting II, we add this new experiment to quantify the impact of excluding parallel data. We upsample the rest of training data to keep the total training volume unchanged, and all other configurations are identical to Setting II.

Results are shown in Table 5. Coincidentally, in our setup, the presence or absence of parallel data has only minor effects on monolingual and cross-lingual abilities, whereas translation performance improves substantially more than the other two categories. This indicates that the inclusion of parallel data primarily contributes to translation ability, aligning with our earlier conclusions.

## 6 CONCLUSION

In this paper, we investigate the role of parallel data in pretraining multilingual large language models. Across a broad range of configurations, we compare the standard practice of using concatenated translation pairs with an alternative that treats each side of the pair as an independent sample, and we obtain consistent findings: parallel data primarily enhances translation ability, but does not directly improve general monolingual competence or general cross-lingual capabilities.

Guided by this insight, we propose to leverage the translation capability induced by parallel data to synthesize targeted corpora to strengthen general multilingual performance, and validate the effectiveness of this approach. Our experiments clarify both the effect and the limits of parallel data in multilingual pretraining, offering actionable guidance and inspiration for building more universally capable multilingual language models.

## 7 LIMITATIONS

While our experiments validate the effects of parallel data across multiple configuration dimensions, they are constrained by computational resources and do not exhaust all possibilities. We did not explore model scales larger than 8B. In terms of data composition, we adopt a setting where English monolingual data accounts for half of the corpus to reflect its practical dominance, but we do not study the impact of varying the English proportion. Our proposed approach leverages translation capability from parallel data to synthesize training corpora. The improvement can depend on the amount and quality of English data source for translation, which we do not analyze. Nevertheless, we note that in practice, high-quality English data is far more available than non-English data, which lends our findings continued practical relevance.

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
