# OpenReview forum: "From Translation to Multilinguality: Revisit the Role of Parallel Data in Multilingual LLM Pretraining"
_ICLR.cc/2026/Conference — ICLR 2026 Conference Withdrawn Submission_

### Official Review · Reviewer_XjaE · 2025-10-15

**Soundness:** 3
**Presentation:** 3
**Contribution:** 3
**Rating:** 6
**Confidence:** 4

**Summary:**

The presented paper presents a thorough study on synthetic multilingual parallel data for LLM training. The authors find that parallel data is mostly beneficial for translation capabilities and does not generalize well to other multilingual benchmarks. Additionally, they propose a framework that boosts general monolingual and cross-lingual evaluation quality by mixing monolingual, code-switched, and parallel training data.

**Strengths:**

- Nicely presented results on multilingual synthetic parallel data
- Thorough experiments on different settings to incorporate parallel data
- Interesting finding that parallel data is mostly helpful for translation capabilities but not other multilingual benchmarks

**Weaknesses:**

### Weaknesses

- To me it is unclear how "Standard" is concatenated. For an `en-de` parallel document with multiple paragraphs `<en_p1>, <en_p2>, <en_p3>` and `<de_p1>, <de_p2>, <de_p3>` is it `<en_p1>, <en_p2>, <en_p3>, <de_p1>, <de_p2>, <de_p3>` or `<en_p1>, <de_p1>, <en_p2>, <de_p2>, <en_p3>, <de_p3>`, i.e. document-level concatenation or interleaving? Both of these approaches might be worth investigating.
- Flores is a sentence level translation testset, it would be beneficial to see how well the ablations/proposed approach works on document-level translation. My assumption is that the synthetic parallel data is mostly sentence-level which is more aligned with the Flores payloads but if the synthetic data would be document level we would see bigger benefits on broader multilingual benchmarks.
- Section 4 Step 2 includes code-switched data where the English sentence is replaced with its translation with 50% probability. I have a few concerns about this approach specifically A) For some language pairs e.g. `en-ja` there might be severe re-structuring happening within the context of a paragraph i.e. the translation might condense multiple English sentences into a single Japanese sentence or change the order of the English sentences to produce coherent Japanese text. How are these edge cases handled? In theory, this isn't too different than the interleaving approach outlined above? B) If we add code-switched data it increases the models probability to produce code-mixed outputs (which is likely not desired?). Are there edge cases observed where the model outputs code-mixed responses or how can we be sure to avoid this behavior?
- It is unclear where are the gains from Table 3 for General Monolingual Ability & General Cross-lingual Ability are coming from. Is it mainly the code-switched or the monolingual data that was generated? How is the monolingual data different than the split approach? Is the English side discarded?


### Minor Comments

- Figure 1 has cut-off % signs on the right side
- Table 1 typo `FloresTranlate` -> `FloresTranslate`

**Questions:**

Part of weaknesses.

---

### Official Review · Reviewer_QzCn · 2025-11-01

**Soundness:** 3
**Presentation:** 3
**Contribution:** 3
**Rating:** 6
**Confidence:** 3

**Summary:**

This paper investigates the role of parallel data in pre-training multilingual large language models (MLLMs). It systematically compares the standard practice of concatenating translation pairs with treating each side as an independent sample. The experiments reveal that while parallel data significantly improves translation performance, it does not directly enhance general monolingual competence or broader cross-lingual abilities. Building on this insight, the authors propose a practical pipeline that leverages translation capability from parallel data to synthesize targeted corpora, resulting in improved general multilingual performance.

**Strengths:**

- generally easy to follow.
- Extensive experiments across 18 languages and multiple pre-training configurations provide robust and consistent evidence.
- The ablation studies and controlled comparisons between concatenated vs. independent parallel data are thorough and informative.

**Weaknesses:**

While the study is extensive, additional discussion on how the method scales with very large corpora or extremely low-resource languages could be helpful.

**Questions:**

- I am curious to see whether this approach has been tested at the document level; most experiments seem sentence-level. Would document-level evaluation yield different insights?
- Have you tried to increase larger proportion of parallel data > 20%?

---

### Official Review · Reviewer_d1tf · 2025-11-03

**Soundness:** 3
**Presentation:** 3
**Contribution:** 1
**Rating:** 2
**Confidence:** 5

**Summary:**

The paper explores the use of parallel data in language modeling. They:
- Ablate on using parallel data as is
- Suggest that using synthetic monolingual data is more useful
- They train models upto 8B, 300B tokens scale to study this.

**Strengths:**

- Ablations are extensive and at a model scale that is more applicable to modern LMs.
- The main utility of this paper to practitioners is scaling up pretraining with synthetic data. However, this study is somewhat limited. Deepening this study would greatly improve the draft.

**Weaknesses:**

- The novelty of this paper is limited compared to the most Related Works that it cites, and is misses significant related work.
- Kale et al 2021 already show that the parallel data matters less with scale.
- In addition, the writing is unclear and feels rushed.

Missed Related Work:
- Work on Self-supervision in MT (https://arxiv.org/abs/2005.04816) and Translationese introducing biases in evaluation as background for using non-English synthetic data. An older survey with useful pointers: https://www.cfilt.iitb.ac.in/resources/surveys/2024/Survey%20Meet%20SyntheticData%202024.pdf
- Using synthetic monolingual data for LM pretraining is a known technique, Eg. https://arxiv.org/abs/2403.13638 -  Even so, I feel this is the more interesting part of the paper and that this should have been explored further at a smaller scale.

Typos And Presentation:
- DATA PREPAREATION ---> DATA PREPARATION
- Focus on monolingual ability is unclear
- Figure on model proportions can be moved to appendix

**Questions:**

- What does equitable mean in this context (line 115)?
- Do you have results split by language? Which languages improve the most? Does code-mixed data matter? or just using more synthetic data?
- Why not use monolingual data as a baseline?  So far, the experiments mainly tell us that the amount of data in the non-English side matters, and you suggest using back/forward-translation, which would indeed likely increase the amount of data for low resource languages in the benchmarks.

---

### Official Review · Reviewer_Yeax · 2025-11-10

**Soundness:** 2
**Presentation:** 2
**Contribution:** 2
**Rating:** 4
**Confidence:** 4

**Summary:**

This paper studies the gain of using parallel data for multilingual LLMs pretraining and finds that while parallel data gains on translation tasks they don't transfer into gains on other tasks.

**Strengths:**

• A valuable topic to study how parallel data affect the performance of pretrained multilingual LLMs
	• Conducted experiments clearly suggest that (1) parallel data only help translation tasks; (2) standard concatenation of parallel data help translation performance significant more; (3) neither standard concetenation nor split protocol of parallel data can help with tasks other than translation.

**Weaknesses:**

• The proposed remedy approach seem to improve multilingual LLMs performance marginally. More study is needed.
	• The study is limited on parallel data's contribution to training LLMs in two forms only: standard and split. Subtlties regarding finner grain of properties of the training data, such as overlapping of tokens and other statistics might be helpful to understand the  nature of parallel data towards training LLMs more.

**Questions:**

1. Section 3.4, please clarify the evaluation protocol. Any finetuning after the pretraining? Or just using prompt template with the pretrained models for translation and other tasks?
	2. In all tables, please clarifiy the \Delta column: (1) average of the difference on individual taks or target languages, or (2) the difference of average over all tasks or over all target languages?

---

### Note · Authors · 2025-12-06

**Comment:**

The authors are grateful to all reviewers for their thoughtful reviews and insightful feedback. Unfortunately, due to time and resource limitation, we were not able to address the concerns or provide follow-up experiments before the deadline. After careful discussion, we've decided to withdraw this paper. We look forward to re-submit this paper in future conference with refined experiments and analysis.

**Withdrawal Confirmation:**

I have read and agree with the venue's withdrawal policy on behalf of myself and my co-authors.